# Immediate Effect Evaluation of a Robotic Ankle–Foot Orthosis with Customized Algorithm for a Foot Drop Patient: A Quantitative and Qualitative Case Report

**DOI:** 10.3390/ijerph20043745

**Published:** 2023-02-20

**Authors:** Dimas Adiputra, Ully Asfari, Mohd Azizi Abdul Rahman, Ahmad Mukifza Harun

**Affiliations:** 1Electronics Technology and Smart Industry Faculty, Institut Teknologi Telkom Surabaya, Surabaya 60231, Indonesia; 2Information Technology and Business Faculty, Institut Teknologi Telkom Surabaya, Surabaya 60231, Indonesia; 3Engineering Faculty, Universitas Sebelas Maret, Surakarta 57126, Indonesia; 4Malaysia Japan International Institute of Technology, Universiti Teknologi Malaysia, Kuala Lumpur 54100, Malaysia; 5Engineering Faculty, Universiti Malaysia Sabah, Kota Kinabalu 88400, Malaysia

**Keywords:** ankle–foot orthosis, robotic AFO, immediate effect, kinematics, spatiotemporal, interview

## Abstract

This study aims to evaluate the immediate effect of a robotic ankle–foot orthosis developed in previous studies on a foot drop patient. The difference with previous research on AFO evaluation is that this research used a setting based on the patient’s request. The robotic AFO locked the foot position on zero radians during the foot flat until the push-off but generates dorsiflexion with a constant velocity in the swing phase to clear the foot drop. A kinematic and spatiotemporal parameter was observed using the sensors available on the robotic AFO. The robotic successfully assisted the foot drop (positive ankle position of 21.77 degrees during the swing phase and initial contact) with good repeatability (σ^2^ = 0.001). An interview has also conducted to investigate the qualitative response of the patient. The interview result reveals not only the usefulness of the robotic AFO in assisting the foot drop but also some improvement notes for future studies. For instance, the necessary improvement of weight and balance and employing ankle velocity references for controlling the walking gait throughout the whole gait cycle.

## 1. Introduction

The human body functions by receiving tons of information from the brain. The brain will malfunction when the blood supply to the brain is interrupted or decreased. This condition is called a stroke. The body’s functionality is reduced because of the stroke, and sometimes it might lead to death. Stroke has existed for a long time as one of the top three causes of death [1]. Despite that, there are a considerable number of survivors, those are to be treated because they often end up as disabled individuals, as reported in [2,3,4]. The treatment is known as stroke rehabilitation.

Rehabilitation aims to recover the body functionality lost due to the stroke through a training scenario. Initially, the brain sends a command to the body part so it can perform accordingly. Because of the stroke, the information path is disrupted; thus, the command cannot reach the targeted body part. A previous study [5] mentioned that moving the body part repeatedly, such as in a training scenario, can ignite a new information path. The more intense the training, the faster the recovery that is expected. Therapist assistance is necessary for the training since the post-stroke patient cannot move their limb well. A routine meeting with the therapist is also essential to ensure the training continuation and to evaluate the patient’s progress until the post-stroke patient recovers the body or limb functionality [6].

The doctor also usually prescribes a wearable device for the patient, such as an orthosis, to assist their body functionality outside the training context [7]. In the case of lower limb rehabilitation, ankle–foot orthosis (AFO) is the most common one. Many researchers have developed ankle–foot orthoses for over four decades [8,9]. Ranging from the most conventional AFO of polypropylene and carbon fiber AFO [10,11] to the most recently developed ones that involves technology, such as robotics [12,13] and electrical stimulation technology [14,15]. The supporting scheme or strategy is varied. The flexible AFO [16,17,18] and passive controlled AFO [19,20,21,22] keep the ankle position at a certain angle to avoid the foot drop [23] or to avoid muscle pain [5,24]. The powered AFO generates ankle movements to force the user to walk normally in several scenarios, such as flat walking [25], sit-to-standing motions [26], and ascending and descending stairs [27]. The Functional Electrical Stimulation (FES) AFO provokes muscle activation by sending electrical signals into the limb accordingly [28]. Here, the AFO prescription should suit the patient’s needs. Under or over-specification is not suggested. For instance, a patient with a severe disability needs a powered or FES AFO to walk, but a patient with a mild disability only needs the passive controlled AFO [29].

Despite many supporting scheme variations, the goal of an AFO remains the same, which is to produce a healthy walking posture for the user during the training session with the therapist and their daily activities. If the healthy gait posture can also be repeated during everyday activities, it is expected to increase the recovery rate [30]. However, another study by [31] also found that the recovery rate is not directly related to training intensification, and moderate-intensity training is preferred over high-intensity. Therefore, the development of an AFO should also focus more on daily activity assistance rather than training intensification [32]. The user’s comfort with AFO usage is essential in designing a suitable AFO. For instance, the following questions must be answered: (i) Is it easy to wear the AFO? (ii) How does the increased weight on the patient’s lower limb due to the AFO affect their tiredness? (iii) How does the AFO usage affect the patient’s confidence? Adding the financial aspect may prevent the answers to these questions from resulting in the most complex and sophisticated AFO.

Studies on the effect of using AFOs cannot be generalized because of a large amount of variation in AFO settings and training scenarios [33]. However, the similarities still exist, such as evaluating the effect of AFO usage immediately [4,17], in the short term [34], or the long term [35]. The immediate effect is significant because it affects the patient’s acceptance of using the AFO for further treatment. Most previous studies implemented controlled variables (i.e., AFO setting/prescription and training setup) according to the researcher’s will. This is not wrong because the setting from the researcher is intended to create the optimum healthy walking gait posture for the user. Although the patient input might contribute to a better set for the AFO, it is still rare that the researcher involves a patient in choosing the AFO settings and training setup. This condition is probably due to the lack of knowledge from the patient about the best settings or prescription.

This study evaluated a robotic AFO that was previously developed in [35]. The robotic AFO is intended for foot drops commonly found in post-stroke patients. The robot has an active actuator in the form of a brushless DC (BLDC) motor, and the controller algorithm was set by accommodating the patient’s request. AFO prescription or setting for the post-stroke patient should be made according to the patient’s needs, which does not allow for under or over-specification of the AFO. Additionally, the user’s comfort is an important parameter, and forcing an AFO setting might be counterproductive. Therefore, the research involved the patient in the process of setting the robotic AFO accordingly. The patient then wore the robotic AFO to perform ground surface walking. The study immediately investigated the patient’s spatiotemporal and ankle kinematics parameters before and after using the robotic AFO. The study examined the patient’s qualitative response before and after using the robotic AFO set according to the patient’s will. Finally, the patient’s reaction after seeing the spatiotemporal and kinematic result when using the robotic AFO was also examined.

## 2. Materials and Methods

### 2.1. Participant

A single patient participated in this study. Although more participants are suggested by previous research to get more general results [16,17], a single participant was suitable for testing the concept offered by the researchers [18,35]. In this case, the provided concept is about the immediate effect of a robotic AFO with customized algorithm on a foot drop patient. The patient details are as follows: male, 56 years old, a weight of 78 kg, and height of 163 cm. He has never used any robotic assistive device in post-stroke recovery. The patient had an ischemic stroke in July 2021 after about one week after contracting COVID-19. In general, the patient’s condition was healthy, but he complained more about the foot drop that made him unable to walk smoothly. Currently, the patient wears a flexible AFO to minimize the difficulty in walking, customized according to his anatomy and condition.

The results of the manual muscle testing (MMT) measurement showed that the patient’s left leg could move with the full range of motion, fight gravity, and withstand maximum loads. In the right leg, the patient was detected to resist full joint motion but could not defy gravity. Measurements of MMT and ROM that are carried out to obtain data from several movements of ankle dorsi flexion, ankle plantar flexion, ankle inversion, and ankle eversion. ROM measurements on the patient’s feet were also carried out on the patient’s left leg for the ankle dorsi flexion, and the eversion ankle could move 20 degrees freely. The ankle plantar flexion and ankle inversion on the patient’s left leg could move 10 degrees freely. However, in the patient’s right leg, the movement of the ankle dorsi flexion was 45 degrees, in contrast to his left leg. On the right foot, the patient’s plantar flexion ankle and ankle eversion could move by 20 degrees, while the inversion of the ankle was the same as the patient’s left leg, which was 10 degrees. In summary, the spasticity only occurred on the right foot with an Ashworth scale of 3 and MMT scale of 2, while the left foot was still fine and could move appropriately without foot drop symptoms.

### 2.2. Customized Robotic AFO

The robotic AFO is a powered AFO with one Degree-of-Freedom developed in [36], and is shown in Figure 1a. It has a brushless DC (BLDC) motor right at the ankle, enabling the robot to flex in the dorsi and plantar direction with a constant speed. An encoder and current sensor are available inside the BLDC motor, which allows ankle position, ankle velocity, and robotic torque measurements. There are also footswitches in the form of a force-sensing resistor (FSR) beneath the insole to detect the gait phases. A microcontroller that controls the motor is placed inside the control box at the back of the calf. A battery is inside the control box to power the robotic AFO, making it a wearable device. The structure was custom-made according to the participant’s limb size, and Velcro straps firmly attached the limb to the robot.

The robotic AFO can classify the walking gait into four gait phases using the FSR sensor: phase 1 (P1) from initial contact to foot flat, phase 2 (P2) from foot flat to heel off, phase 3 (P3) from heel off to toe-off, and phase 4 (P4) from swing phase to the following initial contact [37]. In each phase, the robotic AFO can have a distinguished control algorithm. Initially, the control algorithm was set so that during the swing phase, the robotic AFO moves the foot towards the dorsi direction, and then locks at the desired position. Then, the patient requested an additional function of the robot to lock its position at 0 degrees (the foot and the leg are perpendicular, dorsi is a positive angle, and plantar is a negative angle). The final control algorithm discussed in this study is the middle option: to lock the foot position at 0 degrees in phases 1–3 and lift the ankle to the dorsi direction, then lock it in phase 4, as shown in Figure 1b.

### 2.3. Data Collection

The research collected the data of the patient walking with the robotic AFO on his right leg in two cases: with control and without control. There was expected to be a significantly different response to the foot drop assistance of the robotic AFO [4]. For data collection, the patient walked on a flat surface for about 7–8 steps forwards, turned around, and then walked in the opposite direction for about 7–8 steps. The ankle kinematic and spatiotemporal parameters were measured using the built-in encoder and footswitches. The measured ankle kinematic parameters are the ankle position in one step, ankle velocity in one step, peak plantarflexion angle in loading response, peak dorsiflexion angle in the late stance phase, and ankle ROM in the stance phase. The measured spatiotemporal parameters are the number of steps (cadence) and stance–swing percentage in one step. The number of steps is calculated by finding the average step duration in seconds, then converting it to steps/min. The stance–swing percentage is the ratio between the P1–P2 and P3–P4 duration.

This study also used qualitative methods to explore the needs and conditions of the patient. Since the robotic AFO is in the prototype development phase, a customer approach is the correct way for this research. The qualitative method was chosen because of several considerations, such as:-The optimal robotic AFO setting is still unknown, so it is necessary to gather further patient information.-To understand the meaning of the data that researchers from previous studies have obtained, we tested/practiced these results directly on the patient to determine whether there is a gap between theory and practice.-Understanding the social interactions and feelings between researchers and patients, is to understand the motivation and enthusiasm of both parties to achieve the same goal.-The data used in the development of the robotic AFO no longer comes from case studies in theory but is also reinforced by data originating from a patient who experienced a stroke.

Several questions were asked during an interview, which can be classified into two categories: the satisfaction of using the robotic AFO and about moving using the robotic AFO [35]. The questions included:Satisfaction category:
How do you feel when wearing the AFO Robot?Are you comfortable standing when using the AFO Robot?Are you comfortable sitting when using the AFO Robot?Does the AFO Robot affect your balance?Do you use energy efficiently while using the AFO Robot?Is the AFO Robot comfortable to use?Are you satisfied with the appearance of the AFO Robot?Does the sound of the AFO robot bother you?Will your footwear remain intact when using the AFO Robot?Does the AFO robot cover stay intact?Can you wear different shoes while using the AFO Robot?Are you free while using the AFO Robot?Regarding moving using the robotic AFO:
Can you walk on a flat floor?Are you able to sit and stand from the chair?Can you walk on a slippery floor?


Open-ended questions (i.e., “any suggestion for the robotic AFO?”) were also asked so the patient could request the need to configure the robotic AFO settings.

This interview was repeated in three stages: (1) before using the robotic AFO, (2) after using the robotic AFO, and (3) after seeing the kinematic and spatiotemporal parameters resulting from using the robotic AFO. In the early stages of the discussion, the interviews were conducted based on the questions in the previous chapter. The answers from the patient will become the knowledge to improve and optimize the function of the robotic AFO. The questions were repeated in the second phase confirming the initial request and the robotic AFO outcome. The patient’s kinematic and spatiotemporal data were recorded if the robotic AFO felt comfortable in a walking session. In the final phase, the same questions were asked again as well as showing the patient’s gait measurement that has been recorded by the robotic AFO.

### 2.4. Data Processing

Data processing involves several activities, including separating the whole data into step data, normalizing the step data, calculating the averages of the step data, then checking the variance, as shown in Figure 2. The gait phase data were separated into step data, where one step starts from the initial contact and ends on the following initial contact. When the data was split, they were normalized from step duration to step percentage (0–100%). The step percentage was set to be 0.1%, so that the data had the same data point. After that, the data averages were calculated to obtain representative data. Checking the variance is essential to see whether the average data represents the normalized data well.

As for the interview data, they were initially processed in each stage individually. This study focused on the difference in each stage rather than the overall qualitative response throughout the study. The second reason is that we would like to use the patient’s response to set the robotic AFO accordingly, which was known after the first stage.

## 3. Results

### 3.1. Spatiotemporal Parameters

The normal gait states in order are P1 (initial contact to foot flat), P2 (foot flat to heel off), P3 (heel off to toe off), then P4 (swing). Because of the foot drop, the gait order is P3-P2-P3-P4 instead of P1-P2-P3-P4. The subject started the gait from P3 instead of P1. Then, the gait usually continued to P2, back to P3, then P4. After using the robotic AFO with active control, the gait can start from P1 again instead of P3. The variances of the gait states also decreased. Previously, the variance was 0.29 but became 0.105 after applying the robotic AFO with active control. Figure 3 compares the gait state before and after the subject used the robotic AFO with active control.

Additionally, notice the shifted timing and duration of the gait states, which can be seen in Table 1. The P2, P3, and P4 timing shifted to the right after the subject used the robotic AFO. The second phase started at 8.26% before applying for assistance but shifted to 10.38% after applying for the robotic AFO assistance. The third and fourth phases shifted from 59.96% to 75.35% and from 67.41% to 81.69%, respectively. Consequently, the P1, P2, and P3 duration became longer after using the robotic AFO, while the duration of P4 became shorter.

The cadence, stance, and swing time are shown in Table 2. The cadence decreased, although not significantly, from 18.028 steps per minute to 17.771 steps per minute. The gait distribution percentage hardly changed. Previously, the stance–swing ratio was 60% to 40%. After the robotic AFO assistance, the stance–swing ratio became 75% to 25%, which was farther than the normal stance–swing ratio of 60% to 40% [38]. The change in the stance–swing ratio occurred because of the shifted timing of the beginning of each gait phase, shown in Figure 3. Despite the change, the variance of the stance–swing ratio was significantly smaller because of the robotic AFO assistance (31.37 to 147.07).

### 3.2. Ankle Kinematics

The ankle kinematics was irreconcilable after the subject used the controlled robotic AFO. The ankle position comparison is shown in Figure 4a. The ankle position tended to stay at a negative angle without the robotic AFO assistance due to the foot drop. The ankle position rose to positive angles at 0.38 rad, with the robotic AFO, especially 0–10% (P1) and 80–100% (P4). The ankle position went up on P4 because of the dorsiflexion generated by the robotic AFO. As a result, the ankle position started at the positive ankle position at the beginning of P1. From P2 until P3, the robotic AFO kept the ankle position around zero, as requested by the patient. This result shows that the toe clearance of the patient was achieved using the robotic AFO.

The ankle velocity result also shows that the robotic AFO aid significantly changed the ankle velocity, as shown in Figure 4b. In general, the ankle velocity was negative during P1 and P3 because flexion in this phase is plantar flexion. During P2 and P4, the ankle velocity was usually positive due to the dorsiflexion. Initially, the ankle velocity without the robotic AFO aid ranged from −1400 degree/s to 2000 degree/s, where the positive ankle velocity was dominant during P2 only. The positive ankle velocity was missing during P4 because of the foot drop. The ankle velocity changed drastically from −230 degree/s to 40 degree/s. The positive ankle velocity can be found in P2 and P4, and the negative ankle velocity can be found in P1 and P3. Not only the ankle velocity magnitude but also the variances hardly decreased when the gait was aided by the robotic AFO, which was 491.631 to 0.141. The controlled ankle position had less variance than the uncontrolled one.

An additional comparison of the kinematic parameters before and after applying the robotic AFO assistance is shown in Table 3. The robotic AFO assistance generally decreased the ankle ROM from 9.4 to 2.46 degrees. During P2, the ankle position started at the loading response and ended at the late stance phase. The peak plantarflexion at loading response changed from −7.22 to 0.97 degrees. Typically, plantarflexion is the negative ankle. However, the peak plantarflexion was positive after the robotic AFO aid because of the control algorithm that locks the ankle position around zero degrees. On the other hand, the peak dorsiflexion in the late stance phase increased from 2.12 to 3.38 degrees. The peak dorsiflexion in the late stance phase improved because the starting point (peak plantarflexion at loading response) was lower before the robotic AFO aid.

### 3.3. Interview

Based on the interview results, improvements were made to support the usefulness of the robotic AFO to the patient. In this section, the results were divided into three stages: before using the robotic AFO, after using the robotic AFO, and after seeing the kinematic and spatiotemporal parameters resulting from using the robotic AFO.

In the initial meeting with the post-stroke patient (first stage), discussions were held regarding initial obstacles in activities, perceived complaints, and the patient’s motivation in healing. The patient initially used a static conventional AFO to aid ankle function, especially dorsiflexion by the tibialis anterior (TA) muscle. Because of the stroke, the signal path that sends commands to the TA muscle was disturbed, which resulted in foot drop, and the patient could not optimally control the walking gait. At the beginning of the meeting, there was an attempt to install the robotic AFO on the patient’s foot. The patient explained that the device had a slightly heavy load because the modules and sensors were installed. However, using the robotic AFO felt comfortable. Regarding the balance, using the AFO Robot took time to adjust it. A squeaky sound produced by the robotic AFO was a little disturbing but not so distressing to the patient’s hearing.

The patient requested the robotic AFO should always lock the foot position. The robotic AFO control algorithm initially did not lock the foot position, except during P4. This situation is uncomfortable because the patient used to have a static AFO that always locks the foot position. After using the robotic AFO with the requested control algorithm (second stage), the patient said that the locking feature was already working well. There were still improvement notes from the patient regarding the walking exercise using the robotic AFO. After surviving the stroke, the patient’s leg tended to move outward, unlike the regular leg. Therefore, there should be some additional mechanism to rotate the hip inside to correct the patient’s leg. Despite that, the patient stated that robotic AFO helped to practice in a relaxed condition (sitting supine) so that the muscles continued working even though the patient was not walking. The patient also said that there needs to be an adjustment of the maximum dorsiflexion angle accordingly.

The device was then adjusted again according to the patient’s feedback. The patient could adjust the maximum dorsiflexion using a mobile app. In the third stage, the requested lock function was fulfilled along with the maximum dorsiflexion adjustment. Because of that, the patient now did not mind the weight of the robotic AFO. In this stage, the kinematic and spatiotemporal data when the patient uses the robotic AFO were explained to the patient. The gait was successfully controlled, and the foot drop was successfully prevented. However, the overall walking gait was still far from normal, where there should be dorsiflexion during P2 and plantarflexion during P3. For future research, the patient is willing to apply different control algorithms that have more suitability with a regular walking gait.

## 4. Discussion

First, the limitation of the research is explained. This research only employed an encoder and FSR sensor on the robotic AFO to collect walking gait data, such as ankle position, ankle velocity, and gait phases. The derivative walking gait data, such as the cadence, stance–swing ratio, and ankle ROM, have also been obtained. Ideally, a gait analyzer should be used to observe more comprehensive walking gait data, such as the knee–hip angle and joint moment [35,38]. Walking gait energy expenditure is also interesting because it can add more justification to the robotic AFO usage [39]. For instance, the robotic AFO successfully aids the user’s walking. However, because the robotic AFO is heavy, the walking gait energy expenditure may increase, which is counterproductive.

The expected immediate effect of using an AFO is that the foot drop can be avoided with good repeatability, which can be observed with the current setup. However, the goal of the present study was to observe the immediate effect on a foot drop patient. The robotic AFO avoided the patient’s foot drop with good repeatability, as seen in Figure 4. The ankle position was positive in P1 and P4 with a variance of 0.001, which shows foot drop prevention with good repeatability that suits the expectations. The foot drop patient tended to stumble when walking due to the foot drop.

Consequently, the walking speed also became faster than it should be. The developed robotic AFO assistance should improve this condition by assisting the foot drop to reduce the risk of stumbling and walking speed. The cadence result shows an improvement regarding this matter, where it decreased from 18.14 steps/min to 17.97 steps/min. The reduction was insignificant, but the improvement was in the right direction.

Another interesting discussion is the stance–swing ratio that changed from 60:40 to 75:25 after the robotic AFO assistance. The typical stance–swing ratio is 60:40, and the subject’s gait phase was already similar to the typical stance–swing ratio. Despite the similarity, the percentage of the P1 was only 8.26%, which is shorter than the normal P1 percentage of 10%. After the robotic AFO assistance, the P1 percentage improved to 10%. However, the P2 percentage significantly increased, which resulted in a total stance phase of 75%. The ankle joint locking caused the added P2 percentage during the gait phases. The leg should dorsiflex and reach the positive ankle angle at the end of P2, but the ankle joint was kept at zero rad instead. The stance–swing ratio will affect the walking stride. The longer the stance phase, the shorter the swing phase. This means that the walking stride will be shorter because the swing time is very low [40].

The robotic AFO has successfully aided the foot drop, where the foot drop prevention can be observed with good repeatability. Despite the accomplishment of the primary function, the overall effect, which is the response to the robotic AFO assistance, is still debatable, such as the improved cadences but worsened stance–swing ratio. The next step is to evaluate whether the effect of robotic AFO assistance is helping the overall walking gait or not. Future studies can consider a complete evaluation using a gait analyzer and walking energy expenditure measurement to answer this question [41].

Successful control algorithms affected the patient’s acceptance of the robotic AFO. The interview result told a story where the patient was initially concerned about the robot’s AFO weight and balance. The control algorithm, or rather the way the robot should have functioned, was also a concern. Throughout the process, the robotic AFO weight was not the issue anymore, especially after the patient got the requested functionality from the robotic AFO. Despite that, the issue of the robot’s weight and balance can be optimized from the beginning so that patient’s acceptance can be accelerated. The weight of the robotic AFO might also be the cause of the lower swing ratio. In a sense, the swing cannot be too long because of the robot’s AFO weight.

Education to the patient is also essential. The interview result shows that the patient’s opinion about the robot’s AFO function changed after seeing the kinematic and spatiotemporal results. Initially, the patient wanted the robot to always lock the foot position, like the conventional AFO. Now, the patient is willing to try another control algorithm that can produce a more normal walking gait. In a normal walking gait, the foot position is not always locked. Instead, flexion throughout the walking gait is essential. The robotic AFO has successfully generated dorsiflexion during the P4, which is vital for toe clearance during P1. The dorsiflexion during the P2 and plantarflexion during the P3 are also necessary so the body weight can be shifted forward. This additional functionality should be considered in future studies.

The flexion speed or the ankle velocity of each person might be different. A previous study has shown that ankle velocity is related to the body mass index and walking speed [22,42]. Improvement on the control algorithm should consider this ankle velocity reference in each gait phase because it cannot always be constant. For instance, the plantarflexion speed in P3 is higher than in P1 because a push-off in P3 requires a high force, resulting in high plantarflexion speed. The ankle velocity reference can refer to a healthy ankle velocity [42] to train the walking gait. However, it might not be comfortable for the patient. To make it more comfortable, the robotic AFO could refer to the other healthy legs’ ankle velocity for controlling flexion when walking, especially in daily activities [43].

## 5. Conclusions

This research has presented the immediate effect of robotic AFO usage that has been developed for a foot drop patient. The foot drop has been successfully aided with good repeatability (σ^2^ = 0.001), as shown by the kinematic and spatiotemporal parameters. The interview result revealed not only the usefulness of the robotic AFO in assisting the foot drop but also some improvement notes for future studies. Firstly, the weight and balance can be optimized to accelerate the acceptance of the robotic AFO to the user. Secondly, the robotic AFO should control the flexion with an ankle velocity reference based on a reference obtained from ordinary people walking or the patient’s other leg that is still in healthy condition. Thirdly, a complete gait analysis on the effect of the robotic AFO usage on the overall walking gait experience should be done by using a gait analyzer.

Future studies will also made several customized robotic AFOs for each participant, and they will try the customized algorithm first. Then, after seeing the result, the patient will be asked to use the robotic AFO with a determined algorithm intended to produce a natural walking gait. It will be interesting to see whether the gait is better improved by employing the customized algorithm. Lastly, since the immediate effect of the developed robotic AFO usage on a patient has been proven, then broader population participation is encouraged in a future study to have more significant results in general.

## Figures and Tables

**Figure 1 ijerph-20-03745-f001:**
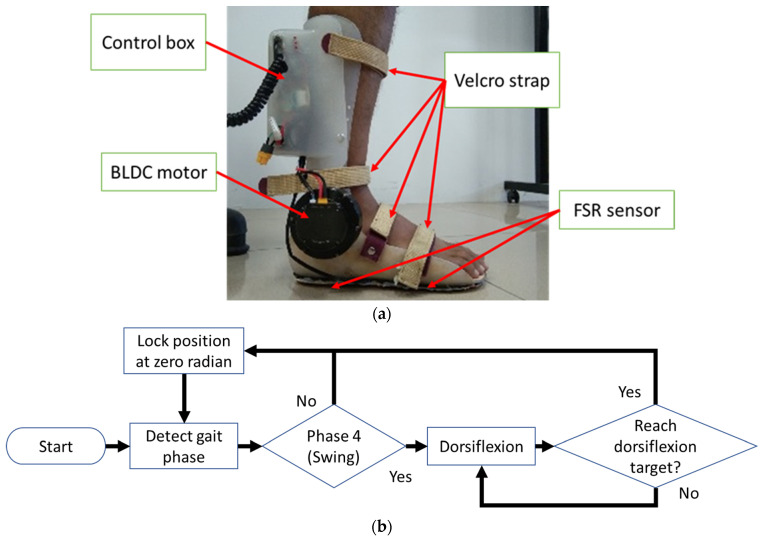
The custom robotic AFO used in this study: (**a**) appearance and (**b**) algorithm flowchart.

**Figure 2 ijerph-20-03745-f002:**
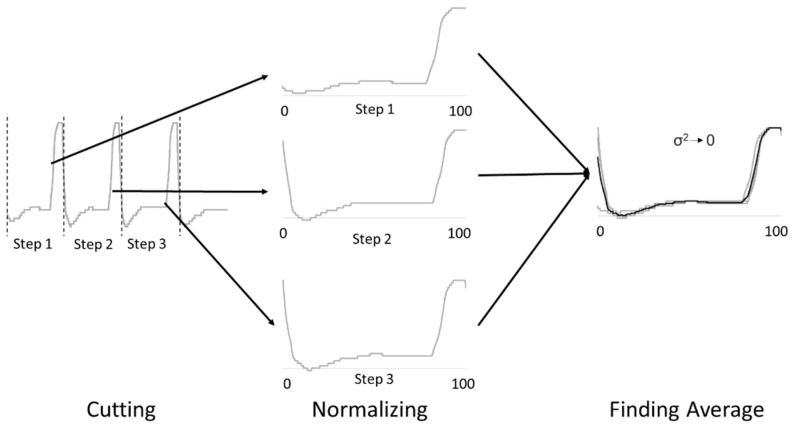
The spatiotemporal and kinematics data processing.

**Figure 3 ijerph-20-03745-f003:**
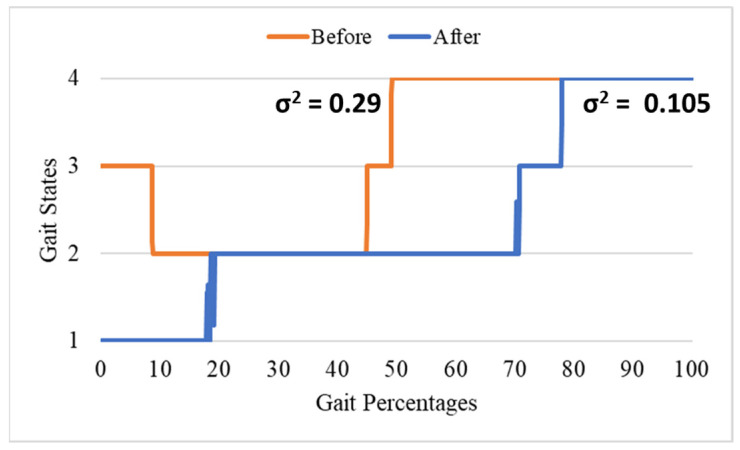
The gait state comparison before and after using the robotic AFO with active control.

**Figure 4 ijerph-20-03745-f004:**
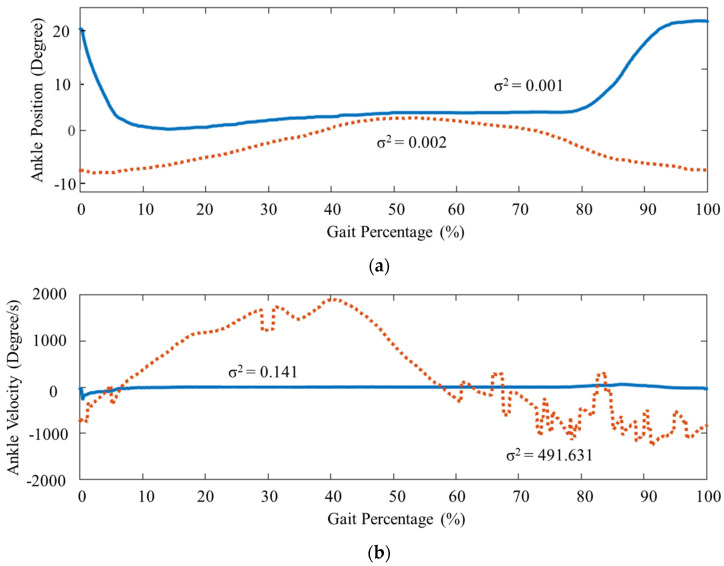
The ankle position (**a**) and ankle velocity (**b**) with and without robotic AFO aid.

**Table 1 ijerph-20-03745-t001:** The starting of each gait phase in gait percentage.

Gait	P1	P2	P3	P4	END
Before	0.00%	8.26%	59.96%	67.41%	100.00%
After	0.00%	10.38%	75.35%	81.69%	100.00%

**Table 2 ijerph-20-03745-t002:** The summary of the spatiotemporal result.

Spatiotemporal Parameter	Unit	Before	σ^2^	After	σ^2^
Cadence	Steps/min	18.14	2.17	17.97	3.52
Stance–swing ratio	Gait %	60:40	147.07	75:25	31.37

**Table 3 ijerph-20-03745-t003:** Kinematic parameter comparison before and after the robotic AFO aid.

Kinematic Parameter	Unit	Before	σ^2^	After	σ^2^
Ankle plantarflexion peak angle at loading response (◦)	Degree	−7.22	0.001	0.97	0.0003
Ankle dorsiflexion peak angle in late stance phase (◦)	Degree	2.12	0.003	3.38	0.000092
Ankle ROM in stance phase (◦)	Degree	9.4	0.004	2.46	0.000545

## Data Availability

The data used in this study is not available publicly. However, researchers who are interested in the data can contact us through email at adimas@ittelkom-sby.ac.id for further discussion or collaboration.

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
