# Peer review of "Immediate Effect Evaluation of a Robotic Ankle–Foot Orthosis with Customized Algorithm for a Foot Drop Patient: A Quantitative and Qualitative Case Report"

_ijerph, 2023, doi:10.3390/ijerph20043745_

Round 1
Reviewer 1 Report
The paper reports a study to evaluate the effect of a robot ankle foot orthosis for a foot drop patient. A single case-study is presented in depth and the manuscript is well-written and interesting, paving the way for future work.
Of course it would be more interesting to see a wider population participating the study, to have more interesting and reliable statistics. This is the major concern about this paper. Nevertheless, the work is remarkable and also single case studies are worth reporting since they are useful to prove the feasibility and interest of what proposed. Therefore, this comment should be intended by the authors as a suggestion to develop their work further and report the next findings also in other papers, so that this one can be seen as one of the first stepping stone into their investigation.
I believe that the authors could improve the introduction by highlighting the importance of single-case studies in the healthcare framework, by citing some recent paper reporting single-case studies.
The manuscript is well written in general, only moderate English changes are required to improve the readability in the whole article.
Author Response
We thank the reviewer for the constructive comment. The followings are the answer to reviewer's comment.
Comment 1: The paper reports a study to evaluate the effect of a robot ankle foot orthosis for a foot drop patient. A single case study is presented in depth, and the manuscript is well-written and interesting, paving the way for future work.
Of course, it would be more interesting to see a wider population participating in the study, to have more interesting and reliable statistics. This is the major concern of this paper. Nevertheless, the work is remarkable, and also single case studies are worth reporting since they are useful to prove the feasibility and interest of what is proposed. Therefore, this comment should be intended by the authors as a suggestion to develop their work further and report the next findings also in other papers so that this one can be seen as one of the first stepping stones into their investigation.
Answer: Thank you for the comment. The conclusion section has added the urgency to see a wider population participating in the study. "Lastly, since the immediate effect of the developed robot AFO usage on a patient has been proven, wider population participation is encouraged in the future study to have more interesting results in general. Several custom robot AFOs will be made for each participant, and they will try the custom-requested algorithm first. Then, after seeing the result, the patient will be asked to use the robot AFO with a determined algorithm intended to produce a natural walking gait. It will be interesting to see whether the gait is better improved by employing the custom requested algorithm or the determined algorithm."
Comment 2: I believe the authors could improve the introduction by highlighting the importance of single-case studies in the healthcare framework by citing recent papers reporting single-case studies.
Answer: Thank you for the comment. The importance of single-case studies has been added in section 2.1 Participant, by citing papers that had been there in this paper, such as (1) J. H. Jeon et al., "Microprocessor‐Controlled Prostheses for a Bilateral Transtibial Amputee with Gait Analysis and Satisfaction: A 1‐Year Followup Case Report," Int J Environ Res Public Health, vol. 19, no. 14, Jul. 2022. and (2) F. M. Kadhim and M. S. Hayal, "Analysis and evaluating of flexible ankle foot orthosis for drop foot deformity," in Defect and Diffusion Forum, 2020, vol. 398 DDF, pp. 41–47.
"A single patient participates in this study. Although more participants are suggested by previous research to get more general results [16, 17], a single participant is suitable for proofing the concept offered by the researchers [18, 35]. In this case, the offered concept is about the immediate effect of a robot AFO with custom requested algorithm on a foot drop patient".
Comment 3: The manuscript is well written in general, and only moderate English changes are required to improve the readability of the whole article.
I believe the revised version of this article has been improved accordingly.
Reviewer 2 Report
The subject of the paper is of great interest in developing and usage original robotic AFO. Your paper represents a useful work in this field. It seems that you have had other contributions in this field and in related domains and have published previous papers.
Please Consider following suggestions:
- Pag. 2, line 84: This study evaluated a robot AFO developed previously in [35]. But reference 35 is J. H. Jeon et al., “Microprocessor-Controlled Prostheses for a Bilateral Transtibial Amputee with Gait Analysis and Satisfaction: A 1-Year Followup Case Report,” Int J Environ Res Public Health, vol. 19, no. 14, Jul. 2022, doi: 10.3390/ijerph19148279. Please correct, probably the reference is [36]. Moreover, in the text, several times reference is made to [35] instead of [36]
- please give more details about the portability / wearability of the robotic AFO, if it is totally autonomous (in terms of energy, for example), and if can be used at home or only in laboratory;
- the main problem that I have identified refers to the fact that it a single patient participates in this study; please give more details concerning your future research and the extension of the studies, based on the results presented in this paper, but with a larger number of users
- minor English corrections are required.
Author Response
We thank the reviewer for the constructive comment. The followings are our answer to the comments.
Comment 1: The subject of the paper is of great interest in developing and using original robotic AFO. Your paper represents useful work in this field. It seems that you have had other contributions in this field and related domains and have published previous papers.
Please consider the following suggestions:
- Pag. 2, line 84: This study evaluated a robot AFO developed previously in [35]. But reference 35 is J. H. Jeon et al., "Microprocessor-Controlled Prostheses for a Bilateral Transtibial Amputee with Gait Analysis and Satisfaction: A 1-Year Followup Case Report," Int J Environ Res Public Health, vol. 19, no. 14, Jul. 2022, doi: 10.3390/ijerph19148279. Please correct me; probably the reference is [36]. Moreover, in the text, several times reference is made to [35] instead of [36]
Answer: Thank you for the comment and advice. The said references have been checked and fixed.
Comment 2: - please give more details about the portability/wearability of the robotic AFO, if it is totally autonomous (in terms of energy, for example), and if it can be used at home or only in the laboratory;
Answer: Thank you for the comment. The robot AFO is a wearable device that operates independently because it has a battery and microcontroller in the control box. This information has been added in sub-section 2.2 Customized Robot AFO.
Comment 3: - the main problem that I have identified refers to the fact that a single patient participates in this study; please give more details concerning your future research and the extension of the studies based on the results presented in this paper, but with a larger number of users
Answer: Thank you for your concern. In the conclusion section, the urgency to see a wider population participating in the study has been added, i.e., "Lastly, since the immediate effect of the developed robot AFO usage on a patient has been proven, then wider population participation is encouraged in the future study to have more interesting result in general. Several custom robot AFOs will be made for each participant, and they will try the custom-requested algorithm first. Then, after seeing the result, the patient will be asked to use the robot AFO with a determined algorithm intended to produce a natural walking gait. It will be interesting to see whether the gait is better improved by employing the custom requested algorithm or the determined algorithm".
Comment 4: - minor English corrections are required.
Answer: Thank you for the comment. English corrections have been fixed.
Reviewer 3 Report
Stroke is a colloquial term for hemorrhagic stroke, a life-threatening brain disorder that requires prompt hospitalization. A stroke may cause high blood pressure, blood clotting disorders, and taking certain medications. One of the first symptoms of the ailment is persistent weakness, muscle paralysis, and paralysis of one half of the body. Stroke is characterized by brain function disorders lasting over a day in the form of muscle weakness or paralysis and sensory disturbances, usually affecting one-half of the body, vision, speech, or gait disorders and coordination. The authors of the publication entitled .... decided to develop an algorithm to improve a patient's rehabilitation process after a stroke. The research topic undertaken in the article is significant. Given the content presented in the article, I have some questions:
a) The present publication's main remark is that it was developed on the example of one patient. Did the authors test the implemented algorithm on other patients?
b) Is the algorithm adapted to build it? i.e., include more limited positions of motion, and is it adapted to other (more severe) cases of post-stroke limb inertia?
c) Is the presented algorithm patented? I suggest the authors because the solution is interesting.
d) Minor editing notes. I wonder if it would be easier for the reader to understand the graphs in Figure 4 if the angular units were expressed not in radians but in degrees. In addition, it would be worth adding a discussion chapter. Including additional literature items related to the presented research topic
Considering the publication, it takes up an interesting research topic. After considering the corrections I suggested, it will be accepted.
Author Response
We thank the reviewer for the constructive comment. The followings are our answer to the comments.
Comment 1: Stroke is a colloquial term for hemorrhagic stroke, a life-threatening brain disorder that requires prompt hospitalization. A stroke may cause high blood pressure, blood clotting disorders, and taking certain medications. One of the first symptoms of the ailment is persistent weakness, muscle paralysis, and paralysis of one half of the body. Stroke is characterized by brain function disorders lasting over a day in the form of muscle weakness or paralysis and sensory disturbances, usually affecting one-half of the body, vision, speech, or gait disorders and coordination. The authors of the publication entitled .... decided to develop an algorithm to improve a patient's rehabilitation process after a stroke. The research topic undertaken in the article is significant. Given the content presented in the article, I have some questions:
- a) The present publication's main remark is that it was developed on the example of one patient. Did the authors test the implemented algorithm on other patients?
Answer: Thank you for the comment. The study has proven the concept of robot AFO usage only to a single patient. Implementation on other patients will be considered in the future by extending the research to have the targeted patients using the robot AFO device with a determined algorithm. The significance of letting the patient customizes the robot AFO control algorithm will be observed. This explanation has been added to the conclusion section.
Comment 2:
- b) Is the algorithm adapted to build it? i.e., include more limited positions of motion, and is it adapted to other (more severe) cases of post-stroke limb inertia?
Answer: Currently, the algorithm is limited to walking motion only. Since it is active-controlled, it will also be suitable for post-stroke patients with spasticity.
Comment 3:
- c) Is the presented algorithm patented? I suggest the authors because the solution is interesting.
Answer: Thank you for the comment and kind advice. The algorithm presented in this study is about actively controlling the ankle angular velocity and is soon to be patented. However, the algorithm to control the ankle angular velocity passively has been patented in Indonesia.
Comment 4:
- d) Minor editing notes. I wonder if it would be easier for the reader to understand the graphs in Figure 4 if the angular units were expressed not in radians but in degrees. In addition, it would be worth adding a discussion chapter, including additional literature items related to the presented research topic.
Answer: The graph has been changed to a degree format. Also, a discussion chapter is already there in the manuscript.
Comment 5: Considering the publication, it takes up an interesting research topic. After considering the corrections I suggested, it will be accepted.
Answer: Thank you very much for your kind support.